# Neonatal Vitamin C and Cysteine Deficiencies Program Adult Hepatic Glutathione and Specific Activities of Glucokinase, Phosphofructokinase, and Acetyl-CoA Carboxylase in Guinea Pigs’ Livers

**DOI:** 10.3390/antiox10060953

**Published:** 2021-06-12

**Authors:** Vitor Teixeira, Ibrahim Mohamed, Jean-Claude Lavoie

**Affiliations:** 1Department of Nutrition, University of Montreal, and CHU-Sainte-Justine Research Centre, 3175 Chemin de la Côte-Sainte-Catherine, Montréal, QC H3T 1C5, Canada; vitor.teixeira.nascimento@umontreal.ca; 2Department of Pediatrics-Neonatology, CHU Sainte-Justine, University of Montreal, 3175 Chemin de la Côte-Sainte-Catherine, Montréal, QC H3T 1C5, Canada; ibrahim.mohamed@umontreal.ca

**Keywords:** glutathione, oxidative stress, vitamin C, cysteine, energy metabolism, metabolic reprogramming, DOHaD, guinea pig, neonate, non-alcoholic fatty liver disease

## Abstract

Premature neonates are submitted to an early-life oxidative stress from parenteral nutrition, which is vitamin C (VC) deficient and induces low endogenous levels of glutathione. The oxidative stress caused by these deficiencies may permanently affect liver glycolysis and lipogenesis. This study evaluates the short- and long-term effects of neonatal VC and cysteine deficient diets on redox and energy metabolism. Three-day-old Hartley guinea pigs from both sexes were given a regular or a deficient diet (VC, cysteine, or both) until week 1 of life. Half of the animals were sacrificed at this age, while the other half ate a complete diet until 12 weeks. Liver glutathione and the activity and protein levels of glucokinase, phosphofructokinase, and acetyl-CoA-carboxylase were measured. Statistics: factorial ANOVA (5% threshold). At 1 week, all deficient diets decreased glutathione and the protein levels of glucokinase and phosphofructokinase, while cysteine deficiency decreased acetyl-CoA-carboxylase levels. A similar enzyme level was observed in control animals at 12 weeks. At this age, VC deficiency decreased glutathione, while cysteine increased it. Acetyl-CoA-carboxylase protein levels were increased, which decreased its specific activity. Early-life VC and cysteine deficiencies induce neonatal oxidative stress and an adult-like metabolism, while predisposing to increased lipogenic rates during adulthood.

## 1. Introduction

Right after birth, oxygen concentrations rise sharply from 20–25 mm Hg to 100 mm Hg, allowing mitochondrial biogenesis and oxidative phosphorylation, with a consequent increase in the generation of reactive oxygen species (ROS) [1,2,3,4]. This oxidative stress early in life is necessary to trigger the transition from fetal metabolism to neonatal and eventually adult metabolism. However, this oxidative stress can be exacerbated by medical interventions at early age, such as supplemental oxygen and parenteral nutrition (PN) that is given to premature infants [5]. This exacerbated early-life oxidative stress has been shown to contribute to the development of several neonatal comorbidities [5,6,7], but also to program health and disease later in life. Premature newborns develop, at adulthood, a higher risk of cardiovascular disease and hypertension [8,9,10], non-alcoholic fatty liver disease [11], and obesity [12].

Neonatal PN is a mode of nutrition administered to premature newborns who have not yet completed gastrointestinal maturity. This PN, although essential for the survival of extremely preterm newborns, is contaminated with peroxides autogenerated in solution. When photoexcited, dissolved riboflavin catalyzes the electron transfer from reducing molecules to dissolved oxygen, in a reaction where the former become oxidized and oxygen is reduced to hydrogen peroxide. Among the reducing molecules oxidized in this reaction, vitamin C is the main contributor, given its antioxidant properties, tendency to lose electrons, and its high concentration in PN solutions [13,14]. Because of this reaction, vitamin C in PN is completely depleted after a few hours [13]. This inadequate infusion is responsible for a decrease of 50% of liver vitamin C concentrations, as shown in an animal model [15]. Simultaneously, peroxides rise to 350 µM in PN solutions and can reach 1500 µM with increasing concentrations of vitamins [14]. These peroxides in PN deplete tissue glutathione and generate oxidative stress [16].

We have previously shown that guinea pigs submitted to PN during the first days of life develop glucose intolerance, are less physically active [17], and have altered levels of key energy metabolism enzymes in the liver at adult life [16]. On this same model of PN-induced oxidative stress, most of early life impacts of PN are due to peroxides, but many long-term effects of PN remain after the effect of early peroxides is prevented. This suggested that factors in PN other than peroxides were the cause of those effects [16]. Vitamin C deficiency in PN solutions and lower glutathione levels are suspected. For instance, early-life PN induces in adulthood lower levels of glycolytic enzymes glucokinase (GCK) and phosphofructokinase-1 (PFK) in the liver, while increasing the levels of acetyl-CoA-carboxylase (ACC), the enzyme that controls de novo lipogenesis [16]. Given that these enzymes control major metabolic pathways that are associated with the development of metabolic diseases, correcting vitamin C and glutathione deficiencies could be a key strategy to decrease the long-term impact of early-life PN.

Ascorbate, or vitamin C, is an essential nutrient for primates and guinea pigs [18], and it is implicated in several redox reactions as a direct antioxidant. It can act as a direct redox cofactor, by recycling oxidized vitamin E and proteins [19], while also participating in epigenetic modulation by its role in the demethylation of DNA and histones [20,21], and oxygen sensing by promoting HIF-1α degradation [22]. Meanwhile, reduced glutathione (GSH) is the most abundant intracellular antioxidant with a high antioxidant capacity (Eº = −240 mV), and it is implicated in antioxidant enzymatic reactions, such as the ones of glutathione peroxidase and glutaredoxin [23,24,25], and its deficiency can activate inflammation and antioxidant pathways via NF-κB and Nrf2 [26,27]. GSH synthesis is controlled by cysteine concentrations, the cellular availability of which is recognized as being limiting for GSH synthesis. [28]. 

GSH and ascorbate spare each other, mainly because GSH can recycle dehydroascorbate back to ascorbate [29,30]. Despite their interconnection, their specific deficiency is expected to have different effects. Thus, this study aims to determine the short- and long-term effects of neonatal vitamin C and glutathione deficiencies on hepatic energy metabolism. These deficiencies were induced by a diet without ascorbate or without cysteine and low in methionine during the first week of the guinea pig’s life. They could exacerbate the physiological oxidative stress necessary for metabolic transition and accelerate it. We hypothesize that an exacerbated oxidative stress, caused by one of these deficiencies, triggers an accelerated metabolic transition in the liver, programming an adult metabolism phenotype at neonatal age in the liver and increasing the risk of metabolic syndrome at adult age. In this study, we induced these deficiencies by complete exclusion of vitamin C or cysteine from the neonatal diet. Cysteine, as the glutathione limiting substrate, was removed from the diet to induce glutathione deficiency.

## 2. Materials and Methods

### 2.1. Materials

All chemical products were purchased from Sigma-Aldrich, Fischer Scientific, Roche Diagnostics, Bio-Rad laboratories, or MP Biomedicals, unless otherwise stated.

### 2.2. Animal Model

Three-day old male and female Hartley guinea pigs from Charles River Laboratories (St-Constant, QC, Canada) were assigned to one of the following groups, each one containing 16 animals (8 males and 8 females) to receive a diet: (1) control (C); (2) vitamin C deficient (VCD); (3) cysteine deficient (CD); or (4) double deficient (DD). After 4 days receiving these diets, from day 3 to 7 days (1 week) of life, half of the animals in each group were sacrificed, and the other half had their diets changed to a long-term standard diet (2041-Teklad Global High Fiber Guinea Pig Diet; Envigo, Madison, WI, USA), which contained adequate amounts of vitamin C and cysteine. 

Cysteine deficient diets also had lower levels of methionine to reduce in vivo transformation of methionine into cysteine, while still providing adequate amounts of methionine for physiological functions [31]. To provide isoproteic diets, the amino acid content of the cysteine devoid diets was balanced with non-essential amino acids for guinea pigs (Ala, Asn, Asp, Glu, Gly, Pro, and Ser). The composition of these diets is described in Table 1. All custom diets were produced by Envigo. Animals were housed in standard conditions in a 12/12 h light/dark cycle throughout the study.

Food Intake was measured at 1 week, and 4, 8, and 12 weeks by the weight difference of feeders placed in cages after 24 h. Since two animals from the same group and sex shared the same cage, the food intake is reported by cage and not by animal in order to avoid distortions in variability. 

Spontaneous physical activity was measured at week 5 by placing animals individually in a cage crossed by infrared beams (Digiscan DMicro Monitor; Accuscan Instruments, Inc., Columbus, OH, USA), as described previously [17]. Each time the animal crosses one of the beams, one beam cut was counted. Counts were registered during 20 min, after 40 min of acclimation.

Glutathione, ascorbate, and dehydroascorbate (DHA). A measure of 250 mg of freshly collected liver was homogenized in 5 volumes of 5% (*w/v*) metaphosphoric acid and centrifuged at 7200× *g*/3 min; the supernatant and the pellet were stored at −80 °C. The reduced (GSH) and oxidized form (GSSG) of glutathione as well as ascorbate and dehydroascorbate in supernatants were separated in an Agilent 7100 Capillary Electrophoresis System (Agilent Technologies, Mississauga, ON, Canada) in a boric acid 200 mM and acetonitrile 20% *v/v* buffer, pH 9.6 [32]. Samples were diluted to reach 1% metaphosphoric acid concentration, and species were separated in a buffer for 10 min after 8 sec sample injection under 30 kV. GSH and GSSG were quantified by absorbance at 192 nm, whereas ascorbate was quantified by absorbance at 268 nm (Appendix A). Samples were treated with DTT 50 mM in order to detect DHA by its reduction to ascorbate (Appendix A). Appendix A shows the relationship between ascorbate and DHA levels. The increase in the ascorbate peak was calculated and DHA levels were extrapolated through the ascorbate standard curve. Total proteins were measured in pellets by the Bradford method using albumin for the standard curve, as previously described [33]. Data are expressed in nmol/mg protein. The redox potential of glutathione was obtained according to the Nernst equation using the molar concentrations of GSH and GSSG, assuming a liver density of one.

Glutathione peroxidase (GPx) and glutathione reductase (GR) assay. Twenty milligrams of liver were homogenized in 9 volumes of TE buffer (50 mM Tris-HCl pH 7.6, 0.1 mM EDTA-Na_2_) and centrifuged at 7200× *g*/1 min. The supernatant was diluted in TE buffer (1:10) and 12.5 µg of protein was used to assay each enzyme. GPx activity was measured in an assay containing 2 mM GSH, 1 mM tert-butyl-hydroperoxide, 0.1 mM NADPH, and 0.1 U/mL of GSSG-reductase diluted in TE buffer. The reaction was started by the addition of the sample. NADPH consumption was measured by its spectrophotometric absorbance at 340 nm for 6 min. GR activity was measured in TE buffer containing 1 mM GSSG and 0.1 mM NADPH. The reaction was started by the addition of the sample and the consumption of NADPH was measured at 340 nm. NADPH concentrations for both assays were extrapolated from a slope calculated with increasing NADPH quantities, ranging from 0 to 1 mmol. 

Glucokinase (GCK) activity was assayed as described previously [16]. Briefly, liver was homogenized in 2 volumes of buffer (100 mM Tris-HCl pH 7.5, 5 mM EDTA-Na_2_, 5 mM MgCl_2_, 150 mM KCl, and 50µM 2-mercaptoethanol) and centrifuged at 1400× *g*/20 min/4 °C. Glucokinase activity was determined by increasing the absorbance at 340 nm in a reaction containing 0.5 mM ATP, 0.4 mM NADP, 0.067 U/mL G6PDH, and 0.5 mM or 100 mM of glucose added to the buffer (100 mM Tris-HCl pH 7.5, 5 mM MgCl_2_) for 1 h/30 °C. Glucokinase activity was calculated by the subtraction of the activity measured with 0.5 mM glucose (hexokinase), from the one measured at 100 mM (hexokinase + glucokinase). Data are presented as U (nmol of NADPH ⋅ min^−1^ ⋅ mg of protein^−1^).

Phosphofructokinase (PFK) activity was assayed as described previously [16]. Briefly, the activity was measured by the change in absorbance at 340 nm in a reaction system coupled with aldolase, triose phosphate isomerase, and GAPDH. Data are presented as U (nmol of NADPH ⋅ min^−1^ ⋅ mg of protein^−1^).

Acetyl-CoA-carboxylase (ACC) activity was assayed as described previously [16]. Briefly, it was extracted from livers with increasing concentrations of polyethylene-glycol-8000 in an extraction buffer (50 mM Tris-HCl, pH 7.5, 250 mM mannitol, 50 mM NaF, 5 mM Na_2_P_2_O_7_, 1 mM EDTA, 1 mM EGTA, 1 mM PMSF, 4 µg/mL soybean trypsin inhibitor, and 1 mM benzamidine). This technique is the same as described by Kudo [34], but reducing agents were removed to allow observation of possible changes in activity protein oxidation. The extracted ACC was assayed in a buffer containing 60 mM Tris-Acetate pH 7.5, 1 mg/mL BSA, 2 mM ATP, 1 mM acetyl-CoA, 5 mM Mg(CH_3_COO)_2_) for 20 min/37 °C. The reaction was started by the addition of 1.64 mM NaH[^14^C]O_3_ (1.35 µCi of 5 mCi/mmol) and 16.6 mM of NaHCO_3_. The incorporation of ^14^C into malonyl-CoA was measured by scintigraphy. Data are presented as U (nmol of NADPH ⋅ min^−1^ mg of protein^−1^).

Western blots of GCK, PFK, and ACC. Western blot methods were described previously [16,35,36]. Primary antibodies were the following: GCK (rabbit GCK polyclonal antibody ab88056, Abcam Plc, ON, Canada), PFK (mouse PFK-1 monoclonal antibody sc-166722, Santa Cruz Biotechnology, Santa Cruz, CA, USA), ACC (rabbit ACC-1 polyclonal antibody #3662, Cell Signaling Technology, Danvers, MA, USA) at 1:1000 dilution. Vinculin was used as a loading control (mouse VCL monoclonal antibody H00007414-M01, Abnova Corporation, Taipei, Taiwan). Secondary antibodies were goat anti-mouse IgG-HRP antibody HAF007 (R&D Systems, Minneapolis, MN, USA) and goat anti-rabbit IgG-HRP antibody W4011 (Promega, Madison, WI, USA) at 1:2500 dilution.

Western blots of Nrf2, HIF-1α, and NF-κB. Liver samples (100 mg) were homogenized in 1 mL of buffer of 320 mM sucrose, 4 mM HEPES-NaOH, pH 7.4, 200 µM deferoxamine, 10 µg/mL PMSF, and Complete Protease Inhibitor (1 tablet/25 mL) (Roche Applied Science). Samples were centrifuged 600× *g*/10 min/room temperature, which allowed the separation of the pelleted nuclei and the cytosolic supernatant. Cytosolic supernatants were concentrated in a 10 kDa Amicon ultra 0.5 centrifugal filter unit and then diluted in 250 µL of storage buffer (20 mM Tris-HCl, pH 7.4, 2 mM EDTA, 250 mM NaCl, 0.1% Triton X-100, 0.01 mg/mL aprotinin, 0.005 mg/mL leupeptin, 4 mM NaVO_3_, and 0.4 mM PMSF) and stored at −80 °C. The pelleted nuclei were washed in 300 µL of PBS added with 200 µM deferoxamine and 10 µg/mL PMSF and pelleted again (600× *g*/10 min/room temperature). The pellet was resuspended in a lysis buffer (50 mM Tris-HCl, pH7.6, 50 mM NaCl, 1% *w/v* sodium deoxycholate, 0.01% *v/v* Triton X-100, 0.01% *w/v* SDS, 13 mM Na_3_VO_4_, and 200 µM deferoxamine) and agitated for 1 h/4 °C. The nuclear fraction was then aliquoted and stored at −80 °C. Fifty micrograms of cytosolic and nuclear extracts were resolved by SDS-PAGE in 4/8% gels, 120 V/2 h/room temperature in a 25 mM Tris, pH 8.3, 200 mM glycine, and 1% SDS. Proteins were transferred to a PVDF membrane 90 V/2 h/4 °C in 5 mM Tris, 38 mM glycine. Membranes were blocked in 5% *w/v* skim milk in PBS-Tween20 0.2% for 1 h/room temperature. Primary antibodies used were the following: anti-Nrf2 antibody (rabbit Nrf2 monoclonal antibody ab62352, Abcam Plc, ON, Canada) (1:1000), anti-NF-κB (mouse p50 monoclonal antibody sc-8414, Santa Cruz Biotechnology, Santa Cruz, CA, USA) (1:1000), and anti-HIF-1α antibody (rabbit HIF-1α polyclonal antibody ab2185, Abcam Plc) (1:2000). Histone deacetylase 1 was used as a loading control for nuclear extracts (rabbit HDAC1 monoclonal antibody, ab150399, Abcam Plc.) (1:1000). Despite HDAC1 levels having been shown to be affected by ascorbate levels [37] in cancer cells, we found no correlation of liver ascorbate levels in our samples (HDAC1 levels = (−0.1039 ⋅ ascorbate) + 10.708; r^2^ = 0.0038), which made HDAC1 a suitable nuclear control protein. Vinculin (cytoplasmic loading control protein) antibody and secondary antibodies were the same used for GCK, PFK, and ACC.

Statistics. Data are presented as mean ± SEM. ANOVAs were performed using two different models of comparison. To test the effect of each deficiency at 1 or 12 weeks of life, a three-way-ANOVA was performed using vitamin C, cysteine, and sex as independent factors. To test the developmental change, a one-way ANOVA was performed. All variables in each group at 1 week were compared to the same group at 12 weeks, to test the effect of age within each group. The significance threshold was set at 5%.

## 3. Results

### 3.1. Diet Type and Phenotypical Changes

Basal characteristics are described in Table 2. The bodyweight was lower in all deficient groups at one week of life; however, only animals in cysteine deficient groups experienced weight loss. Animals in the VCD group gained weight compared to the control group. Liver weight was not affected at 1 week of age. At 12 weeks of age, the percentage of bodyweight attributed to the liver weight was lower in cysteine deficient animals. Sex was an important factor at 12 weeks, as females had significantly lower bodyweight (F_Sex(1,24)_ = 58.73, *p* < 0.001; F: 521 ± 11 g vs. M: 641 ± 12 g) and liver weight (F_Sex(1,24)_ = 20.35, *p* < 0.001; F: 20 ± 0.8 g vs. M: 16 ± 0.6 g), but this difference was not present when liver weight was normalized by bodyweight, meaning the difference is due to bodyweight differences.

At 1 week of life, the food intake was lower in cysteine deficiency groups (F_Cysteine(1,23)_ = 40.3; *p* < 0.001) (CD and DD) than in C and VCD groups. In function of age, food intake in control and DD groups fitted in a linear model (r^2^_C_ = 0.76, *p* < 0.01; r^2^_DD_ = 0.65, *p* < 0.01), while VCD and CD groups fitted in a quadratic model (r^2^_VCD_ = 0.45, *p* < 0.05; r^2^_CD_ = 0.41, *p* < 0.05), meaning the progression curves were different among groups (Figure 1A). Thus, at 12 weeks of age, food intake was 40% lower in animals having received a deficient diet neonatally (VCD, CD, DD: 31 ± 1.9 vs. C: 51 ± 4.4 g/day; (F_(1,12)_ = 118.65, *p* < 0.001).

Cysteine deficiency groups were associated with a lower level of spontaneous physical activity at 1 month of age (Figure 1B).

### 3.2. Ascorbate and Dehydroascorbate (DHA)

At 1 week of age, liver ascorbate was lower in vitamin C deficiency groups (Figure 2A). At 12 weeks of age, the significant interaction between early-life vitamin C and cysteine deficiencies led to further analyses. Hepatic ascorbate level was higher in the CD group (F_(1,28)_ = 6.22, *p* < 0.05). This increase was not observed in animals that received the DD during their first week of life (Figure 2A). Animal sex was not a significant variable in liver ascorbate or DHA levels. DHA was not affected by any of the deficiencies, neither at 1 week nor at 12 weeks of life (Figure 2B). The mean ratio of DHA on AA was 8.9 ± 0.8%. Similar ratios (5–25% DHA/total ascorbate), determined by high-performance liquid chromatography, have already been reported in guinea pigs’ livers [38,39]. It has been suggested that guinea pigs have a higher recycling capacity of DHA, as they cannot generate ascorbate from glucose like rats and mice. Changes in hepatic ascorbate were not sufficient to influence the levels of HIF-1α (the degradation of which is dependent on ascorbate level) neither in neonatal nor in adulthood (Figure 2C,D).

### 3.3. Glutathione and Oxidative Stress

At 1 week of life, compared to the control group, GSH level was decreased by both VCD and CD. However, their significant interaction suggested that their effects reached a plateau because the value observed in the DD group was not further decreased (Figure 3A). A similar impact of deficiencies on the redox potential of glutathione was observed; both deficiencies led to a plateau with a mean value of −202 ± 2 mV, more oxidized than control −218 ± 3 mV (Figure 3C). Since deficiencies had no impact on GSSG levels (Figure 3B), changes in the redox potential of glutathione appear to be caused by lower levels of GSH. There was no statistically significant difference between groups for nuclear Nrf2 (Figure 3E). However, an interaction between vitamin C and cysteine deficiencies was observed for the cytoplasmic levels of Nrf2 (the stimulation of which favors the synthesis of GSH), blunting the effect of vitamin C and cysteine deficiencies (Figure 3F), much like the effect observed in GSH levels and redox potential. Changes in glutathione were not associated with modification in NF-κB levels, which is normally activated by oxidative stress (Figure 3G,H). The animal sex had no significant impact on glutathione, NF-κB, or Nrf2 levels.

At 12 weeks of life, early-life vitamin C deficiency decreased GSH levels, oxidized redox potential, and decreased nuclear levels of Nrf2 (Figure 3A,C,E). Early-life cysteine deficiency increased both GSH and GSSG levels (Figure 3A,B). Vitamin C and cysteine deficiencies both increased NF-κB cytoplasmic levels, but the addition of their deficiencies in DD group blunted the statistical effect (Figure 3H). GSSG levels, redox potential of glutathione, and nuclear NF-κB levels were higher in female animals, independently of diets (GSSG: + 56%; redox potential: 3.6 mV; Nuclear NF-κB: 46%), whereas cytoplasmic Nrf2 was lower in females, compared to males (−18%).

### 3.4. Glutathione Peroxidase and Reductase

At 1 week of life, vitamin C deficiency increased the activity of GPx by 11% while no difference between groups was observed in the activity of GR (Figure 4A,B). At 12 weeks of age, an interaction between early-life vitamin C and cysteine deficiencies was observed for GPx and GR, as the activities of the enzyme in VCD and CD groups decreased while it increased in DD (Figure 4A,B).

### 3.5. Energy Metabolism Enzymes

At 1 week of life, only the activity of PFK was increased by vitamin C and cysteine deficiencies (Figure 5B). However, specific activities of GCK and PFK were increased by both deficiencies (Figure 5G,H) while that of ACC was increased only in cysteine deficient groups (Figure 5I). These changes in specific activities corresponded to a decrease in protein levels (Figure 5D–F). There was no sex effect on these parameters.

At 12 weeks of life, GCK activity and its protein levels were not affected by any of the deficiencies (Figure 5A,D). PFK activity was higher in males, compared to females (Figure 5B). No effect was observed in its protein levels (Figure 5E), suggesting stimulatory post-translational activity in males but not in females; however, specific activity was not statistically affected (Figure 5H). ACC activity was not affected either (Figure 5C), but since its protein levels increased in all deficient groups (Figure 5F), its specific activity decreased (Figure 5I), which represents an inhibitory post-translational modification. ACC-specific activity was also increased in females (+40%).

### 3.6. Developmental Change

According to the hypothesis of the study, the metabolic maturation is disturbed by early life oxidative stress. Thus, it is important to compare the activity of key enzymes between 1 and 12 weeks of life (Figure 6).

The activity and protein level of GCK (Figure 6A,D) were not statistically affected by age within each group. PFK activity was not significantly changed in the control group, but it was increased in all three deficiency groups (Figure 6B). PFK protein levels (Figure 6E) decreased with age in the control group but remained stable in deficient animals throughout life. ACC activity was decreased in the control group with age, with parallel decreases in deficiency groups (Figure 6C). ACC protein levels (Figure 6F) decreased with age in the control group, did not change in VCD animals, and increased in both CD and DD, showing different patterns from the control group.

## 4. Discussion

The study shows that vitamin C and cysteine deficiencies during the neonatal period share some similarities in their short- and long-term impact in addition to having some specific effects. The results support the initial hypothesis that neonatal oxidative stress can induce an adult-like phenotype of the energy metabolism in the liver of newborns.

At 1 week of life, both deficiencies generated oxidative stress by reducing the levels of GSH and oxidizing the redox potential. Similar oxidation of hepatic redox potential has already been observed in a similar model of neonatal antioxidant deficiencies [35]. It is intuitive that cysteine deficiency would decrease GSH levels, as cysteine is the limiting substrate for GSH synthesis [28,40]. However, vitamin C deficiency also caused a similar decrease in GSH levels, while also decreasing its own levels. Given the difference in their hepatic concentrations, the regeneration of ascorbate from dehydroascorbate [41,42] by GSH does not completely explain the entire decrease in GSH. The increased levels of Nrf2 in deficient animals and the lack of difference in GSSG levels and glutathione reductase activity do not suggest a lower synthetic capacity or recycling of GSH. This surprising observation remains without explanation, but it is of importance to investigate knowing that large parts of the population have suboptimal vitamin C levels [43,44,45,46,47]. This low GSH level explains the oxidation of redox potential, a marker of oxidative stress. The magnitude of the stress was not sufficient to simulate NF-κB but sufficient to stimulate Nrf2, which characterizes this as a mild oxidative stress [48]. These two transcription factors are sensitive to the redox environment. NF-κB is associated with pro-inflammatory cytokine production [49,50,51] and Nrf2 promotes the expression of genes associated with antioxidant defense such as glutathione peroxidase and genes involved in the synthesis of glutathione [52,53].

Despite VCD and CD inducing a similar response on glutathione metabolism at 1 week of life, at 12 weeks they differ. By this age, neonatal vitamin C deficiency was associated with lower levels of GSH (–16%) while neonatal cysteine deficiency was followed by an increased GSH (+21%) levels. Their effects were additive as GSH level in the double deficiency group did not differ from control. Vitamin C deficiency remained without effect on GSSG, while neonatal cysteine deficiency was associated with a greater level of GSSG. Therefore, the redox value in the cysteine deficiency groups remained similar to that of the control group, whereas it was more oxidized in animals with neonatal vitamin C deficiency. The high level of GSH and GSSG in the cysteine deficient group could explain the low level of nuclear Nrf2. However, the low Nrf2 level in the vitamin C deficiency is inconsistent with the low GSH and high redox potential in these animals. The mechanism leading to this low nuclear Nrf2 level in the VCD group remains to be determined.

At 1 week of life, the protein levels of all three enzymes were lower than those of the control group, except for the vitamin C deficiency group for ACC. The absence of change in GCK and ACC activities, and even an increase in PFK activity, in the deficiency groups suggests a post-translational stimulation that counteracts the decrease in protein levels. It is known that an oxidizing environment influences the activity of all these enzymes [16,36,54,55]. For instance, ACC activity is controlled by phosphorylation through the redox-sensitive enzymes, AMPK and PP2A [56]. Therefore, the oxidized redox potential observed in deficient groups could induce inhibition of PP2A, allowing ACC to remain phosphorylated and more active. Eleven weeks after stopping the deficient diets, the values of redox potential and the protein levels and activity of GCK and the activity of ACC were not different among groups.

Prenatal metabolism is characterized by high glycolytic and lipogenic rates, as the lack of oxygen allows energy production from glycolysis and storage of pyruvate as triglycerides. At birth, glycolytic rates slowly decrease and remain low until adulthood as oxygen increases the ATP yield, while lipogenesis plummets at the first days of life and slightly increases at adulthood [57,58]. Comparisons of the values observed at 1 and 12 weeks of life support the hypothesis that oxidative stress can induce an adult-like phenotype of energy metabolism in newborns at the level of protein expression, as the lower levels of glycolytic enzymes found at 1 week of life in deficient animals are similar to the ones observed at adult life in controls. The relationship between the redox potential of glutathione and DNA methylation in the liver of one-week-old guinea pigs has been previously reported. Oxidation of redox potential strongly correlated with increased methylation [59,60], while the methylation of the glucokinase gene promoter was increased with age and associated with the risk of metabolic disease [61,62]. If this relationship could explain our observation, it also suggests that methylation of DNA during neonatal period may not be permanent. Further studies should investigate the hypothesis that a high redox potential promotes higher methylation of genes encoding enzymes of energy metabolism, and that these genes could be demethylated if redox value changes to a lower value.

The particular case of the neonatal cysteine deficiency is interesting. At 12 weeks of age, these animals presented higher PFK activities, which stimulate glycolysis and provide acetyl-CoA, the substrate for ACC. Therefore, even if the activity of ACC, the limiting step in fatty acids synthesis, was without observable change, the increased amount of substrate should result in a greater production of fatty acids in vivo. This is particularly important in an environment where the post-translational repression observed is removed and ACC is more active [63,64]. In brief, the data alert us that neonatal cysteine deficiency may induce a greater risk of fatty liver in adulthood.

Beyond these biochemical changes, a modification in the phenotype was also observable. The fact that after 11 weeks on a normal-standard diet the food intake was 40% lower in animals that received a deficient diet at neonatal age supports an impairment of energy metabolism. However, this large difference did not seem to influence body weight at this age. In a similar animal model, a lower spontaneous physical activity 13 weeks after receiving parenteral nutrition during their first week of life has been observed [17]. Animals from the cysteine deficiency group were also less physically active at 5 weeks of life. Parenteral nutrition also led to a deficiency in glutathione due to a lack of cysteine [17,65]. This spontaneous decline in physical activity in adulthood has been associated with an overall energy deficiency [17].

The initial hypothesis and the aim of the study concerned the impact of neonatal vitamin C and cysteine deficiencies. Sex was not determined to be an important factor for the variables measured in this study at neonatal age. However, we observed that females had increased GSSG and nuclear NF-κB, and an increased ACC specific activity, while also having a decreased PFK activity and cytoplasmic Nrf2 compared to males at 12 weeks of life. These differences seem to be associated with the development of the animals as these are not present early in life and there were no interactions between sex and any of the deficient diets. The increase in nuclear NF-κB seems to be associated with the oxidation of redox potential of glutathione only in females, which has been demonstrated before [27,66,67]. This makes females particularly at risk of developing inflammatory diseases.

This study is one of the few to investigate the effects of neonatal nutrition on programming energy metabolism later in life. The guinea pig is an adequate model to study the developmental origins of health and disease as their perinatal development is similar to humans [68]. The guinea pig is the only mammal along with primates and humans whose vitamin C level can be lowered by a deficient diet because ascorbate is an essential vitamin in these species [18]. Another strength of this study is the use of outbred animals, which allows higher genetic variability and a robust model. In our model, we were limited to the impacts of these nutritional deficiencies in the liver, while we are aware that they could affect energy and redox metabolism in other organs, such as muscle [69], lungs [70], and brain [71]. Another limit of this study is the decrease in food intake in animals receiving a diet deficient in cysteine at 1 week of age. Although these animals were still feeding, they lost weight and it is impossible to determine the contribution of this lower energy intake at neonatal life on the results observed. Although cysteine plays a role in taste [72], and that taste perception is primordial for the food intake and growth of guinea pigs [73], there is no evidence that cysteine or methionine could have an orexigenic effect [74]. This decrease in food intake remains to be confirmed or explained. Our findings create new investigation opportunities to understand the fine biochemical mechanisms linking these nutritional deficiencies to observations.

## 5. Conclusions

To our knowledge, this is the first study to report the programming effects of vitamin C and cysteine deficiency in energy and glutathione metabolism, although many studies have observed the short-term effects of antioxidant deficiencies in guinea pigs [75,76,77,78,79]. The study highlighted the dual importance of preventing these nutritional deficiencies in the newborn, for their immediate health and their health throughout their lifetime. These deficiencies are frequent, particularly in premature newborns under parenteral nutrition. According to the data presented in this study, these deficiencies induce an adult-like phenotype of energy metabolism at neonatal life, while also increasing the risk of fatty liver disease at adult life. Further studies aimed at improving PN formulations in order to prevent these deficiencies are essential, especially in females, who seem to be more responsive to oxidative stress later in life and may develop more inflammation.

## Figures and Tables

**Figure 1 antioxidants-10-00953-f001:**
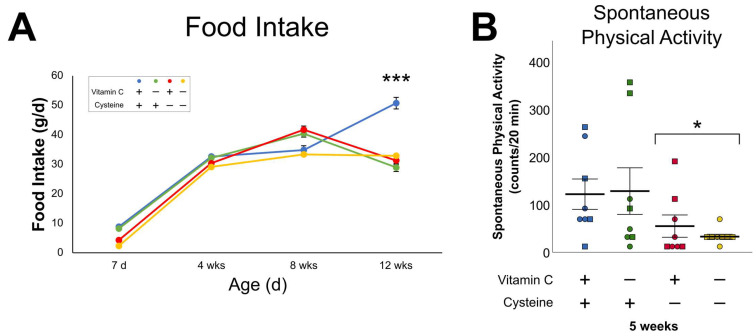
Food intake in animals at weeks 1, 4, 8, and 12 of life and spontaneous physical activity at 5 weeks of life. (**A**): Food intake evolved differently in animals according to the neonatal diets. Controls and double deficient animals evolved linearly (r^2^_C_ = 0.76; r^2^_DD_ =0.65), while vitamin C deficient and cysteine deficient animals fitted in a quadratic model (r^2^_VCD_ =0.45; r^2^_CD_ =0.41). The food intake is significantly decreased in all deficient animals at 12 weeks of age. (**B**) Spontaneous physical activity at 5 weeks was significantly decreased in cysteine deficient animals. Mean ± SEM. Circles: male animals; squares: female animals; *: *p* < 0.05; ***: *p* < 0.001.

**Figure 2 antioxidants-10-00953-f002:**
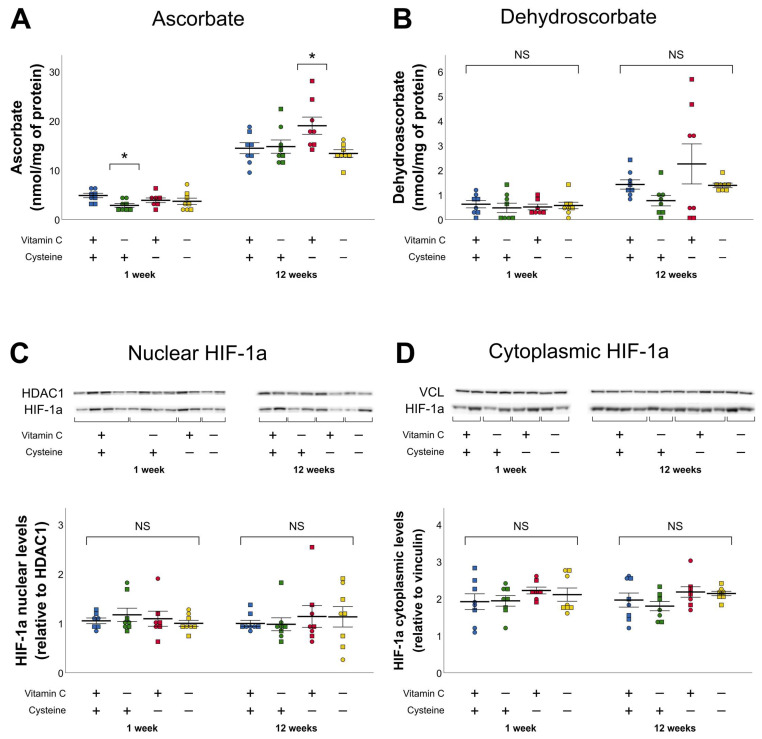
Hepatic levels of ascorbate, dehydroascorbate, and HIF-1α in nucleus and cytoplasm in 1-week- and 12-week-old animals. (**A**). Ascorbate levels were decreased at 1 week of life by vitamin C deficiency, and were increased at 12 weeks of life in the cysteine deficiency group, with a significant interaction between both deficiencies (*p* < 0.05). (**B**–**D**): No statistically significant differences were observed in dehydroascorbate or HIF-1α levels in nucleus or cytoplasm either at 1 week of life or at 12 weeks. Mean ± SEM. Circles: male animals; squares: female animals; NS: statistically non-significant; *: *p* < 0.05.

**Figure 3 antioxidants-10-00953-f003:**
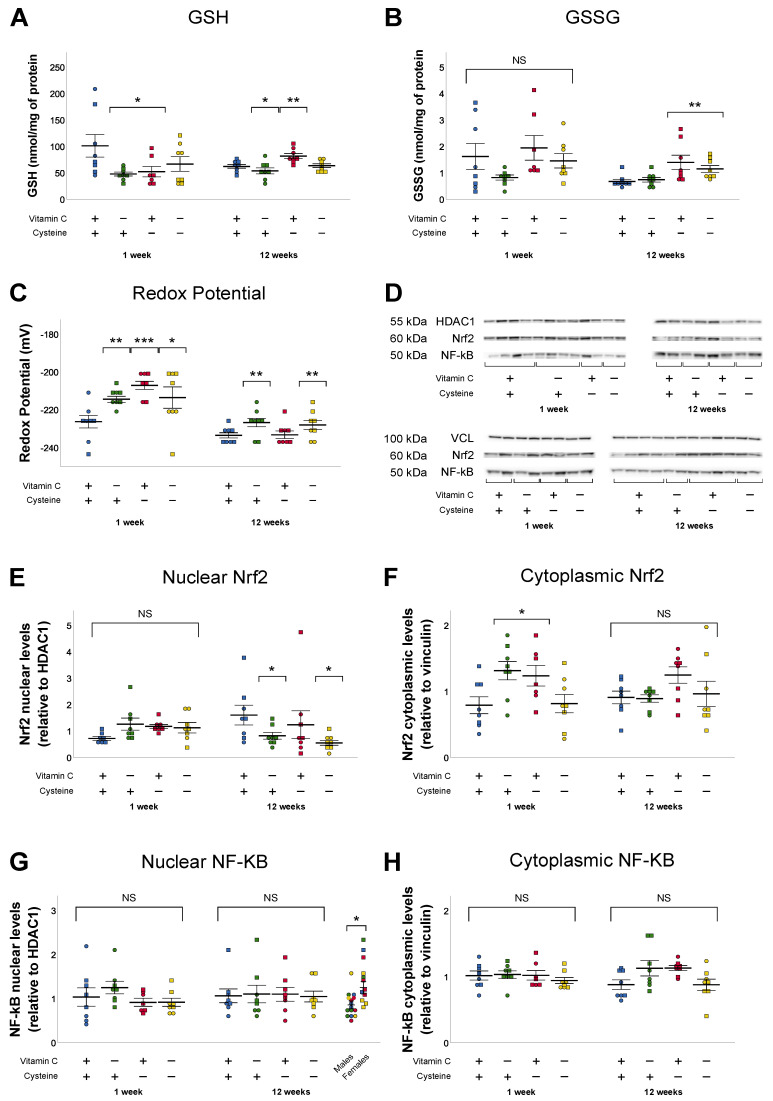
Hepatic levels of GSH, GSSG, redox potential of glutathione, Nrf2, and NF-κB in nucleus and cytoplasm in 1-week- and 12-week-old animals. (**A**). GSH levels were decreased in vitamin C and cysteine deficient animals at 1 week of life. At 12 weeks of life, early-life vitamin C deficiency decreased GSH, while early-life cysteine deficiency increased it. (**B**). At 12 weeks, cysteine deficiency increased GSSG levels. (**C**). The redox potential of glutathione was oxidized (increased) by both deficiencies at 1 week of life. At 12 weeks, redox potential was oxidized by early-life vitamin C deficiency. (**D**). Western blot images from Nrf2 and NF-κB, with HDAC1 and vinculin as reference proteins. (**E**). At 12 weeks of life, early-life vitamin C deficiency decreased nuclear Nrf2. (**F**). Cytoplasmic Nrf2 was increased in both vitamin C and cysteine deficiency animals at 1 week of life. (**G**). Females had increased nuclear NF-κB at 12 weeks of life, compared to males. (**H**). No statistically significant differences were observed in NF-κB levels in nucleus or cytoplasm either at 1 week of life or at 12 weeks. Mean ± SEM. Circles: male animals; squares: female animals; NS: statistically non-significant; *: *p* < 0.05; **: *p* < 0.01; ***: *p* < 0.001.

**Figure 4 antioxidants-10-00953-f004:**
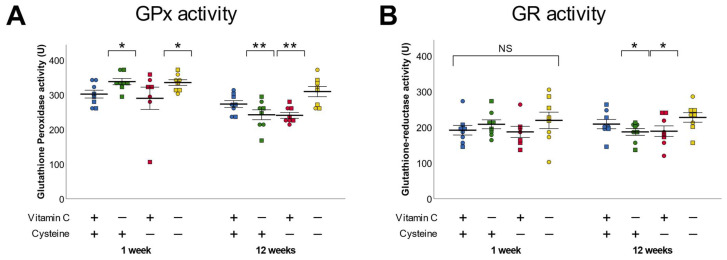
Activity of glutathione peroxidase (GPx) and reductase (GR) in liver of 1-week- and 12-week-old animals. (**A**). GPx activity was increased in vitamin C deficient animals at 1 week of life (*p* < 0.05). At 12 weeks, early-life vitamin C and cysteine single deficiencies decreased GPx activity. (**B**). No statistically significant differences were observed in GR activity at 1 week of life. At 12 weeks, both neonatal vitamin C deficiency and cysteine deficiency decreased GR activity. Mean ± SEM. Circles: male animals; squares: female animals; NS: statistically non-significant; *: *p* < 0.05, **: *p* < 0.01.

**Figure 5 antioxidants-10-00953-f005:**
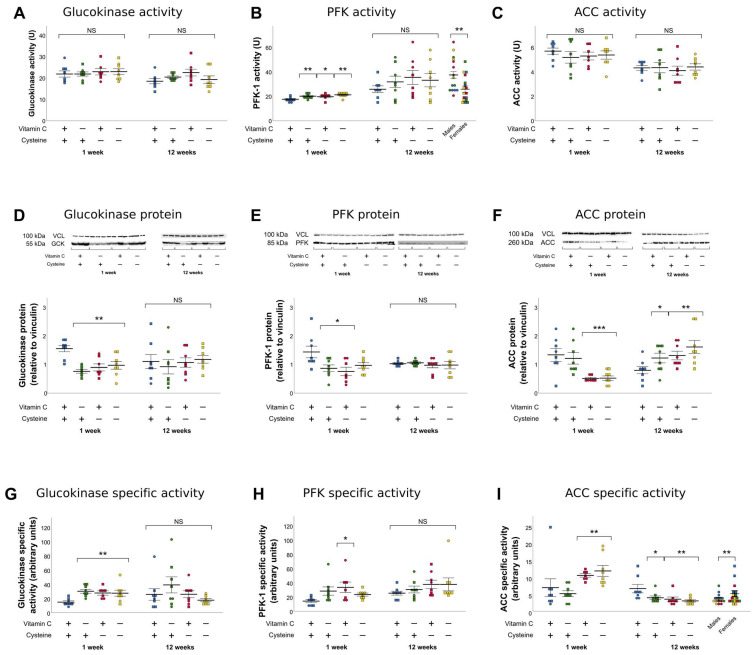
Hepatic activity, protein levels, and specific activity of glucokinase (GCK), phosphofructokinase (PFK), and acetyl-CoA-carboxylase (ACC) in 1-week-old and 12-week-old animals. (**A**). No statistically significant differences were observed in GCK activity either at 1 week of life or at 12 weeks. (**B**). At 1 week of life, PFK activity was significantly increased by vitamin C and cysteine deficiency. At 12 weeks, PFK activity was increased in males, compared to females. (**C**). ACC activity was not significantly affected either at 1 week or at 12 weeks. (**D**). The protein levels of GCK were decreased at 1 week by both vitamin C and cysteine deficiency. (**E**). PFK protein levels were decreased at 1 week by both vitamin C and cysteine deficiencies. (**F**). ACC protein levels at 1 week were decreased by cysteine deficiency. At 12 weeks, ACC protein levels were increased by both early-life vitamin C and cysteine deficiencies, with a further additive effect of the double deficiency. (**G**). At 1 week of life, GCK specific activity was increased by vitamin C and cysteine deficiencies. (**H**). At 1 week of life, PFK specific activity was increased in all deficient groups. (**I**). ACC specific activity at 1 week was increased in cysteine deficient animals. At 12 weeks, it was independently decreased in neonatal vitamin C and cysteine deficient animals, as well as in males. Mean ± SEM. Circles: male animals; squares: female animals; NS: statistically non-significant; *: *p* < 0.05; **: *p* < 0.01; ***: *p* < 0.001.

**Figure 6 antioxidants-10-00953-f006:**
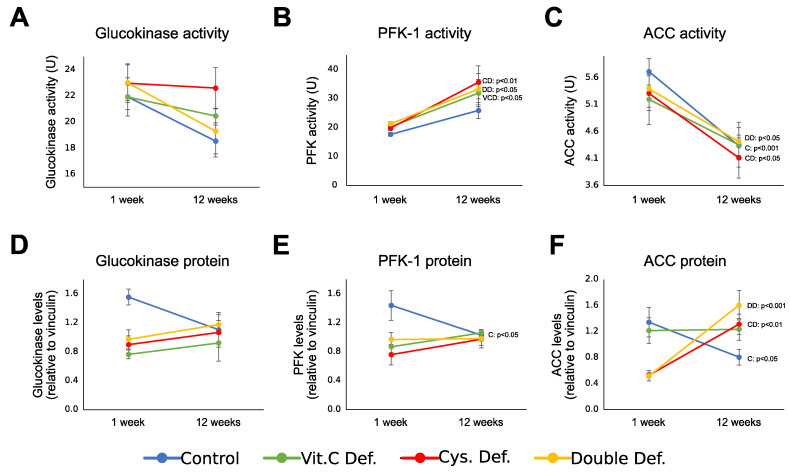
Developmental change of activity and protein levels of GCK, PFK, and ACC in liver of 1-week- and 12-week-old guinea pigs. (**A**). GCK activity did not significantly change with age in any of the groups. (**B**). PFK activity was not significantly increased in the control group with age, but it increased with age in all deficient groups. (**C**). ACC activity significantly decreased with age in control, cysteine deficient, and double deficient groups. (**D**). GCK protein level was not significantly affected by any of the neonatal deficiencies. (**E**). PFK protein levels decreased with age in control animals, but not in deficient groups. (**F**). ACC protein levels decreased with age in control animals, was not affected in the vitamin C deficiency group, and increased with age in both cysteine deficient animals. Mean ± SEM. C: Control group; VCD: vitamin C deficiency group; CD: cysteine deficiency group; DD: double deficiency group.

**Table 1 antioxidants-10-00953-t001:** Nutritional composition of diets used in the study.

	Control Diet (C)	Vitamin C Deficient Diet (VCD)	Cysteine Deficient Diet (CD)	Double Deficiency Diet (DD)	Long-Term Standard Diet
Energy (kcal/g)	3.5	3.5	3.5	3.5	2.4
Protein (% kcal)	18.4	18.4	18.4	18.4	32
Carbohydrate (% kcal)	55.7	55.7	55.7	55.7	18
Fat (% kcal)	25.9	25.9	25.9	25.9	50
L-alanine (g/kg)	3.65	3.65	4.01	4.01	9
L-aspartic acid (g/kg)	3.65	3.65	4.19	4.19	17
L-cystine (g/kg)	2.70	2.70	-	-	3
L-glutamic acid (g/kg)	41.67	41.67	42.27	42.27	27
Glycine (g/kg)	24.45	24.45	24.76	24.76	10
L-methionine (g/kg)	2.90	2.90	2.00	2.00	3
L-proline (g/kg)	3.65	3.65	4.12	4.12	11
L-serine (g/kg)	3.65	3.65	4.08	4.08	10
Vitamin C (mg/kg)	203	-	203	-	1050

**Table 2 antioxidants-10-00953-t002:** Basal characteristics at 1 and 12 weeks of life depending on neonatal diet.

	1-Week-Old Animals				12-Week-Old Animals			
	Control (C)	Vitamin C Deficient (VCD)	Cysteine Deficient (CD)	Double Deficiency (DD)	Control (C)	Vitamin C Deficient (VCD)	Cysteine Deficient (CD)	Double Deficiency (DD)
Bodyweight at day 3 (g)	113 ± 4	100 ± 3	107 ± 3	107 ± 3	107 ± 4	100 ± 3	105 ± 3	99 ± 4
Bodyweight at sacrifice (g)	124 ± 4	**107 ± 4 ****	**104 ± 2 ****	**101 ± 3 ****	577 ± 25	588 ± 34	599 ± 29	558 ± 21
Bodyweight change from day 3 to sacrifice (%)	110 ± 3	107 ± 1	**97 ± 1 *****	**94 ± 1 *****	542 ± 21	594 ± 40	575 ± 37	572 ± 31
Liver weight at sacrifice (g)	3.8 ± 0.4	3.4 ± 0.1	3.2 ± 0.1	3.3 ± 0.1	18 ± 1.1	20 ± 1.7	18 ± 1.1	16 ± 0.7
Liver weight/bodyweight (%)	30 ± 2	32 ± 1	31 ± 1	33 ± 1	31 ± 1	34 ± 2	**29 ± 1 ****	**30 ± 1 ****

Data are reported as mean ± S.E.M. (*n* = 8); **: *p* < 0.01; ***: *p* < 0.001.

## Data Availability

Not applicable.

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
