# Peer review of "Neonatal Vitamin C and Cysteine Deficiencies Program Adult Hepatic Glutathione and Specific Activities of Glucokinase, Phosphofructokinase, and Acetyl-CoA Carboxylase in Guinea Pigs’ Livers"

_antioxidants, 2021, doi:10.3390/antiox10060953_

Round 1

Reviewer 1 Report

Comments  for the authors:

In the present study, the authors examine the hepatic metabolic consequences in adult animals after neonatal vitamin C and cystein deficiencies. Presented results  are interesting, useful and can have an important clinical message and practical implications. I have one main comment to this study and a few minor issues.

First of all, the authors should add another Table with the basal metabolic parameters such as body and liver weight, serum glucose and lipids, adipose tissue weight, e.g, that can appropriately supplement the results observed.

Minor comments:

  • In conclusion, it would be appropriate to mention the differences between male and female. In my opinion, these results are interesting and I think, it is a pity that they are not part of the conclusion.
  • It would also be useful to enlarge all figures that are difficult to read.
  • In the method section, in my opinion, the exact type of statistical method should be mentioned.

Author Response

Answers (A) to the Comments (C) from Reviewer 1:

C: First of all, the authors should add another Table with the basal metabolic parameters such as body and liver weight, serum glucose and lipids, adipose tissue weight, e.g, that can appropriately supplement the results observed.

A: Thank you for this comment. The requested table was added to the manuscript after the first paragraph of the result section. The table contains initial bodyweight, bodyweight at sacrifice, liver weight at sacrifice and percentage of liver weight to whole bodyweight. Unfortunately, we did not collect data on fat pad weight because we have not found any significant differences in fat pat weight in our model of neonatal parenteral nutrition (Teixeira V et al. J Dev Orig Health Dis. 2021 Jun;12(3):484-495). This nutrition mode provokes much more drastic changes, as it induces vitamin C deficiency, glutathione deficiency and it is contaminated with peroxides, that also induce oxidative stress. Therefore, we decided not to collect data on fat pad weights. The same is true for plasma glucose.

Minor comments:

C: In conclusion, it would be appropriate to mention the differences between male and female. In my opinion, these results are interesting, and I think, it is a pity that they are not part of the conclusion.

A: Indeed, the differences found are very interesting as they seem to confirm what is observed in literature for most species (Klein SL, Flanagan KL. Nat Rev Immunol 2016, 10, 626–38; Dunn SE et al. J Exp Med 2007, 2, 321–30.). We have not added these discussions on the first version of the manuscript as these results were secondary and the manuscript was already very dense in information. We developed the paragraph that discusses the sex differences in the discussion section (penultimate paragraph of the discussion section), and we integrated these results into the conclusion (page 18 of the Revised manuscript with track changes). Sex differences were also added in the result section and in the Figure 5.

C: It would also be useful to enlarge all figures that are difficult to read.

A: The figures were enlarged and rearranged for a better visualisation and comprehension.

C: In the method section, in my opinion, the exact type of statistical method should be mentioned.

A: We have rewritten the statistical analysis section to make it clear what kind of method was used. We have changed the word “ANOVA” for “three-way-ANOVA” and “one-way-ANOVA”, accordingly (page 6).

Thank you for your feedback. You really helped us improve our presentation.

Reviewer 2 Report

The paper entitled “Neonatal vitamin C and cysteine deficiencies program adult hepatic glutathione and specific activities of glucokinase, phosphofructokinase and acetyl-CoA carboxylase in guinea pigs’ liver” by Teixera et al compared the long-term impact of postnatal parenteral food deficient in vitamin C and Cysteine on several hepatic proteins and hepatic mechanisms of guinea pigs.

The subject is within the scope of the journal. It represents a large amount of work aiming at testing the hypothesis that early oxidative stress leads to metabolic changes. The observed changes are small but apparently significant. The paper is moderately well written as it contains very long paragraph containing highly speculative aspects of limited use. The figures are numerous and could be improved by adding some titles (e.g nuclear Nrf1 vs cytoplasmic Nrf2). The absence of impact of several tested markers should also be integrated in the discussion.

General.

The study is biased to a very limited set of proteins and assays. At the omics area, such study appears obsolete. To my opinion the use of the animals would be more relevant for RNAseq, proteomics or even redox proteomics. It is very likely that such unbiased approach may decipher novel molecular pathways. Moreover, a large part of the study is based on Western blots, a notoriously non-quantitative method.

Minor.

Abstract, lane 27: enzyme phenotype: find a more suitable term

Lane 38: reactive oxygen species instead of reactive oxidant system

Table 1: is it L-cystine or L-cysteine: double check

Change 7 days to week 1 at several places (Food intake, Fig. 1A,…)

Add molecular weight of each protein tested in western blot

Author Response

Answers (A) to the Comments (C) from Reviewer 2:

C: The subject is within the scope of the journal. It represents a large amount of work aiming at testing the hypothesis that early oxidative stress leads to metabolic changes. The observed changes are small but apparently significant. The paper is moderately well written as it contains very long paragraph containing highly speculative aspects of limited use. The figures are numerous and could be improved by adding some titles (e.g nuclear Nrf1 vs cytoplasmic Nrf2). The absence of impact of several tested markers should also be integrated in the discussion.

            A: We agree that the discussion has long paragraphs. We trimmed the discussion to the most relevant aspects of the study, while also integrating the absence of impact of some of the tested markers. The images were enlarged and adjusted to make the variable names more visible.

C: The study is biased to a very limited set of proteins and assays. At the omics area, such study appears obsolete. To my opinion the use of the animals would be more relevant for RNAseq, proteomics or even redox proteomics. It is very likely that such unbiased approach may decipher novel molecular pathways. Moreover, a large part of the study is based on Western blots, a notoriously non-quantitative method.

            A: We use a translational hypothesis-driven approach for our studies. This means we test physiologically and pathologically possible mechanisms. As much as this approach may limit the potential of discovery of new markers, it also focuses the research on what is relevant for the direct application of the study results in the clinical setting. We agree the use of animals would be relevant for a global approach, and this is indeed one of the next steps of this work. . This is the meaning of the last sentence of the last paragraph (in the new version) of the discussion: Our findings create new investigation opportunities to understand the fine biochemical mechanisms linking these nutritional deficiencies to observations. We intend to get a global metabolomic profile of these animals in order to be able to identify newborns that were more affected by these deficiencies early in life and treat them as early as possible. However, the work is essential before a metabolomics work, in order to establish the long-term effect of these deficiencies in redox and energy metabolism.

Minor

C: Abstract, lane 27: enzyme phenotype: find a more suitable term

A: Lane 26: “Enzyme phenotype” was replaced by “enzyme level”.

C: Lane 38: reactive oxygen species instead of reactive oxidant system

A: Lane 37: “reactive oxidant species” was replaced by “reactive oxygen species”.

C: Table 1: is it L-cystine or L-cysteine: double check

            A: The diets use L-cystine and not L-cysteine. L-cystine is more stable for long-term storage of the diets and it does not confer any advantage or disadvantage over use of cysteine in the diet. The sodium-independent transporter LAT2 that exchanges neutral amino acids like cysteine and cystine has similar affinity for both forms, and once transported inside, cystine is reduced to cysteine, therefore making the use of any of these forms interchangeable (Kohlmeier, M. Amino Acids and Nitrogen Compounds. Nutr Metab 2015, 265–477).

C: Change 7 days to week 1 at several places (Food intake, Fig. 1A,…)

A: “7 days” was replaced by “week 1” in line 6 of the Abstract, line 5 of the Animal model paragraph in page 3, Food intake page 4 and in legend of Figure 1A page 8.

C: Add molecular weight of each protein tested in western blot

            A: The molecular weight of all proteins tested in western blots were added in figures 2C, 2D, 3D, 5D, 5E and 5F. The reference protein was added at the top in all figures, regardless of the molecular weight of the analysed protein, for standardisation and easiness to read.

Thank you for your feedback. You really helped us improve our presentation.

Reviewer 3 Report

There a number of concerns about this manuscript as written, and some additional data is needed to validate some of the experiments taken. Also, the formatting of the data/figures and associated text makes the manuscript hard to read. It will need revision and further rounds of review before publication. A limited list of issues to be addressed follows:

  1. In the methods, are there any references for the use of the capillary electrophoresis system for ascorbic acid? In particular, the pH of the buffer system seems incompatible with the stability of ascorbate. Were the ascorbate standards also assessed for dehydroascorbate content?
  2. Related to question #1, the absorbance maxima of ascorbate is affected by pH. What is the extinction coefficient for ascorbate used at 258nm - please provide a reference for this value at this pH? Can you confirm that the ascorbate absorbance maxima is at 258 at this pH?
  3. Also related to the dehydroascorbate quantification - can you confirm that the DHA absorption at 258nm (in blanks) has been subtracted from these data?
  4. Was the ascorbate status of the pregnant dams assessed? What diet were these animals maintained on?
  5. In the results, starting at the first paragraph of 3.1, the text mixes the results of the initial weight gain with the adult weights. This makes it very hard to follow. Please move the text dealing with early age together at the beginning and move the text concerning the adult animals together in a later paragraph.
  6. In the graphs, the mean appears to be represented as an X, but this X is not easy to see in many of the figures, especially since the error bars are larger than anything else. This needs to be adjusted so that means are clearly seen on the graph. Also, there should be a way of noting significant differences on the graph near the appropriate columns, instead of reading it in the text.
  7. In Figure 2B, there is no way of relating dehydroascorbic acid content to ascorbate content in the liver - is there any relationship between these values?
  8. Dehydroascorbate content in the liver is typically negligible, the pH of the buffer system used (see question 1-3) brings into doubt the validity of the data in Figure 2B. Additional data is needed to clarify the data in this figure to make it acceptable for publication.
  9. For Figure 2C: In some publications, HDAC1 has been suggested as being responsive to ascorbate status. Are there any additional data or references that suggest that liver HDAC1 is stable in these animals in relation to liver ascorbate content?
  10. The figure legends in this manuscript are very lengthy. These should be trimmed and the relevant text brought into the results. Figures 3 and 5 are particularly hard to read or see on the page - there may be too much data here for these figures.
  11. The rationale for the in-depth investigation into cellular metabolic enzymes in these animals is unclear. Are there any indications that mitochondria function or other markers of energy balance are perturbed in these animals?
  12. The authors claim that the glutathione data presented shows an increase in oxidative stress by the reduction of glutathione - this is not accurate. There is no indication of oxidative stress or damage in this system - only a reduction in antioxidant protection. If a statement about oxidative stress is to be made, the authors would need to show an increase in oxidation or a direct response to oxidant production.

Author Response

Answers (A) to the Comments (C) from Reviewer 3:

C: In the methods, are there any references for the use of the capillary electrophoresis system for ascorbic acid? In particular, the pH of the buffer system seems incompatible with the stability of ascorbate. Were the ascorbate standards also assessed for dehydroascorbate content?

A: The method used was similaranother publication where the method was developped (Ref: Dresler, S.; Maksymiec, W. Acta Sci Pol Hortorum Cultus 2013, 6, 143–55.); this reference was added in themethod section page 4. Indeed, the basic pH of the resolving buffer is not compatible with the range in which ascorbate is stable (around pH 3.5). However, it is important to keep in mind that the samples are extracted and storaged in metaphosphoric acid 5%, which keeps pH low and ascorbate and glutathione conserved. The analytes are resolved in an alkaline buffer (pH 9.6) for 10 min, which, considering the short time of migration, (5 min for ascorbate), it should not be enough time to degrade ascorbate into dehydroascorbate. We added an image in supplemental materials (Supplemental Figure 2) that depict the blanks and three concentrations (5, 10 and 20 µM) of an ascorbate standard. It can be seen that there are no contaminating peaks that could be dehydroascorbate.

C: Related to question #1, the absorbance maxima of ascorbate is affected by pH. What is the extinction coefficient for ascorbate used at 258nm - please provide a reference for this value at this pH? Can you confirm that the ascorbate absorbance maxima is at 258 at this pH?

            A: Thank you for your comment as it allows us to correct a typing mistake in the text. The determination was done at 268 nm. We added an image in supplemental materials (Supplemental Figure 1) depicting the absorbance maxima of ascorbate. The image shows the light spectrum of absorbance (200 to 600 nm) of an ascorbate standard at 20µM stored in metaphosphoric acid 5% and resolved in boric acid 200mM/Acetonitrile 20%, pH 9.6. The precise wavelength of absorbance maxima was 268 nm. According to Witmer JR et al (Witmer, J.R.; Wetherell, B.J.; Wagner, B.A.; Du, J.; Cullen, J.J.; Buettner, G.R. Redox Biol 2016, 8, 298-304), the molar extinction coefficient for ascorbate in plasma is 13,000 M-1 cm-1 at 265 nm. This data has not been incorporated into the manuscript.

C: Also related to the dehydroascorbate quantification - can you confirm that the DHA absorption at 258nm (in blanks) has been subtracted from these data?

            A: Dehydroascorbate was quantified by the increase in ascorbate peak area after samples were treated with dithiothreitol (DTT). We added an image to supplemental materials showing the increase in ascorbate peak area due to reduction of dehydroascorbate into ascorbate. This image concerns the liver of the animal with the highest concentration of DHA we found (5.7 nmol/mg of protein) in CD group at 12 weeks, for illustrative reasons. Since DHA is not estimated in a separate peak, and that there are no other peaks that could correspond to DHA in chromatopherograms in AA standards nor blanks the subtraction is not necessary. However, we understand how this method can cause confusion on how DHA was quantified. Therefore, we have re-written the methods in order to avoid confusion (page 4).

C: Was the ascorbate status of the pregnant dams assessed? What diet were these animals maintained on?

            A: The dams are housed in Charles River Laboratories, and we buy newborn guinea pigs for the study, so the vitamin C status of dams could not be assessed. According to Charles River Laboratories, the dams were fed a regular diet originally devoid of ascorbate which is added of ascorbic acid by spraying of a 10% ascorbate solution, giving the diet about 1071 mg of ascorbate/kg of diet, a value similar to our long-term standard diet (1050 mg/kg). According to Charles River Laboratories, the animals do not develop any sign of vitamin C deficiency.

C: In the results, starting at the first paragraph of 3.1, the text mixes the results of the initial weight gain with the adult weights. This makes it very hard to follow. Please move the text dealing with early age together at the beginning and move the text concerning the adult animals together in a later paragraph.

            A: the paragraph was re-written, and it starts with 1-week animals results and is followed by the ones at 12 weeks. It is now at the beginning of the 3.1 section, page 6.

C: In the graphs, the mean appears to be represented as an X, but this X is not easy to see in many of the figures, especially since the error bars are larger than anything else. This needs to be adjusted so that means are clearly seen on the graph. Also, there should be a way of noting significant differences on the graph near the appropriate columns, instead of reading it in the text.

            A: The X in graphs was replaced by a thick horizontal line that is larger than error bars, making it easier to see. Lines in each of the groups are close to the average line on the neighboring group, which makes comparisons between groups easier and more illustrative. The significant differences are shown in the graphs by stars, instead of text.

C: In Figure 2B, there is no way of relating dehydroascorbic acid content to ascorbate content in the liver - is there any relationship between these values?

            A: The relationship between ascorbate and dehydroascorbate is inexistent in 1-week-old animals (r2=0.0018; DHA=((-0.0115 • [ascorbate]) + 0.5881nmol/mg prot), but very linear in 12-week-old animals (DHA = (0.2395 • [ascorbate]) - 2.2368 nmol/mg prot; = 0.5902; p<0.001). It is impossible to determine if the linearity is dependent solely on age, or if it is a result of the effects of the nutritional deficiencies during the first week of life. These relationships are depicted in the Supplemental Figure 3.

C: Dehydroascorbate content in the liver is typically negligible, the pH of the buffer system used (see question 1-3) brings into doubt the validity of the data in Figure 2B. Additional data is needed to clarify the data in this figure to make it acceptable for publication.

A: Indeed, the concentrations of DHA in liver can be lower than in other organs. We have added before de last sentence of the paragraph 3.2: “The mean ratio of DHA on AA was 8.9 ± 0.81 %. Similar ratios (5-25% DHA/total ascorbate), determined by high-performance liquid chromatography, have already been reported in guinea pigs’ livers (Cui Y., et al J Nutr Sci Vitaminol (Tokyo) 2001, 4, 316–20/ Frikke-Schmidt H et al. Redox Biol 2016, 8–13). It has been suggested that guinea pigs have a higher recycling capacity of DHA, as they cannot generate ascorbate from glucose like rats and mice.”

C: For Figure 2C: In some publications, HDAC1 has been suggested as being responsive to ascorbate status. Are there any additional data or references that suggest that liver HDAC1 is stable in these animals in relation to liver ascorbate content?

            A: Indeed, there are a few reports showing the impact of ascorbate supplement on increasing levels of HDAC1 (Mustafi S, et al. EBioMedicine. 2019 May; 43:201-210.). The use of HDAC1 in our study comes from the difficulty of 1) finding a nuclear housekeeping protein with a molecular weight that is not so close to our measured proteins; 2) that is not affected by any of the treatments and 3) that the protein has commercially available antibodies that match the guinea pig sequence, as most of the antibodies are produced for the rat, mice, and human proteins. We tested tubulin, TATA-box binding protein (TBP) and HDAC1. Tubulin antibodies did not show any signal, meaning they probably did not recognize the guinea pig protein. We were able to find a signal with TBP, but it seemed to be increased by the deficiencies by 60% at 1 week of age and decreased by 40% at 12 weeks of age. The last tested protein, HDAC1, gave us a smaller and accepted variation (around 10%) between controls and deficient animals. HDAC1 levels were not correlated with liver ascorbate either (HDAC1 levels = (-0.1039 • Liver ascorbate) + 10.708; = 0.0038). Therefore, we decided to use HDAC1 as a reference protein for our western blot experiences. We added a sentence in the method section (page 6) “Western Blots of Nrf2; HIF-1a; NF-kB” western blot method section that mentions the lack of correlation between ascorbate and HDAC1 levels.

C: The figure legends in this manuscript are very lengthy. These should be trimmed and the relevant text brought into the results. Figures 3 and 5 are particularly hard to read or see on the page - there may be too much data here for these figures.

            A: We agree with your comment. All figure legends were shortened. Non-significant data, p-values and color-coding were removed from legends, as these are already depicted in the images. This reduced the legends in figures 3 and 5 specifically by half.

C: The rationale for the in-depth investigation into cellular metabolic enzymes in these animals is unclear. Are there any indications that mitochondria function or other markers of energy balance are perturbed in these animals?

A: Glutathione deficiency is known to cause mitochondrial and cellular damage in guinea pigs and newborn rats (1), while maternal malnutrition programs higher PGC-1a, a mitochondrial coordinator, in rats (2). Prematurity in itself causes mitochondrial dysfunction, leading to lower brain ATP in rats (3). Even though we do not have evidence that early-life ascorbate and cysteine deficiencies will have an impact on mitochondrial function, the gatekeeper hypothesis, combined with evidence from other models of nutritional insults, suggests mitochondria should be affected in our animals too. This hypothesis states that “there are a limited number of genes or gene pathways that are influenced by a nutritional insult and that changes to these core biological processes represent the primary response to insult from which all subsequent events leading to disease are derived.” (4). No changes to the text are made.

  1. Meister A. Biochim Biophys Acta [Internet]. 1995 May 24 [cited 2019 Nov 26];1271(1):35–42.
  2. Theys N et al. J Nutr Biochem [Internet]. 2011;22(10):985–94.
  3. Sánchez-Alvarez R, Almeida A, Medina JM. Pediatr Res. 2002;51(1):34–9.
  4. McMullen S, et al. Med Hypotheses [Internet]. 2012;78(1):88–94.

C: The authors claim that the glutathione data presented shows an increase in oxidative stress by the reduction of glutathione - this is not accurate. There is no indication of oxidative stress or damage in this system - only a reduction in antioxidant protection. If a statement about oxidative stress is to be made, the authors would need to show an increase in oxidation or a direct response to oxidant production.

A: This is a very relevant comment. Our intervention is considered very mild, as we do not induce any oxidant factors in the biological system, but we deprive animals from antioxidant molecules. These deprivations can be compensated by activation of other pathways, at least in part. In addition, these deficient diets are only provided for 4 days. Therefore, the animals are expected to have a mild oxidative stress. There are several levels of oxidative stress that are described here (Lushchak, V.I. Free radicals, reactive oxygen species, oxidative stress and its classification. Chem Biol Interact 2014,164–75). The oxidative stress we present here is the first levels of oxidative stress, as we observe a decrease of antioxidant defences (glutathione) and an activation of Nrf2, in order to compensate for this decrease. The second level is where the oxidative stress will generate physiological damage, and it is characterized by the activation of NF-kB, AP-1 and MAP kinases that will trigger an inflammatory response and will reprogram other cellular functions. Our data suggest this level of oxidative stress has not been reached as NF-kB does not seem to be activated. The third and last level of oxidative stress, is reached when the cell cannot cope with oxidative stress and enters in apoptotic or necrotic death. We have already found a high level of oxidative stress on our parenteral nutrition model in guinea pigs’ lungs, where we found the activation of Nrf2 and NF-kB, associated with an increase in caspase-3 activation (apoptosis) and a lower alveolarization index (Elremaly W, et al. Redox Biol. 2014 May 20;2:725-31).

This rationale was integrated into the discussion of results in the second paragraph, so it is clear for the readership that the oxidative stress induced by the vitamin C and cysteine deficiencies is low grade oxidative stress in the liver.

Thank you for your feedback. You really helped us improve our presentation.

Round 2

Reviewer 1 Report

All comments were accepted, I recommend to publish.

Good luck. 

Reviewer 2 Report

The manuscript is substantially improved and reads much better. Although I understand the hypothesis-driven approach, I still believe that broader omics approaches may provide additional relevant unsuspected findings. This is particularly relevant in the context of  animal experiments and the 3R (reduce, refine, reduce) rules, which should be applied as much as possible by biomedical scientists.

Reviewer 3 Report

This paper is improved and the author's comments are satisfactory.